# A First-Time Investigation into Ecofriendly and Biocompatible Mg-Se Binary System for a Greener Earth

**Michael Johanes** [ID]**, Vasuudhaa Sonawane and Manoj Gupta** *[ID]

Department of Mechanical Engineering, National University of Singapore, 9 Engineering Drive 1, Singapore 117575, Singapore
* Correspondence: mpegm@nus.edu.sg

**Abstract:** In this study, the Mg-15Se binary system was, for the first time, investigated and synthesized using the powder metallurgy (PM) method, including microwave sintering and hot extrusion. The resulting material was shown to possess visible pores with a porosity of 2.91%, higher than other Mg materials synthesized using this method in the literature. Despite this, the material not only exhibited a comparable corrosion response with pure Mg but also a significantly superior mechanical response (76% greater damping capacity, 57% increase in hardness, and increases of 21%, 50%, and 51% for compressive yield strength, ultimate compressive strength, and fracture strain, respectively). Thus, this not only opens the door for future work concerning the addition of medicinal Se to nutritional Mg element and the optimization of process parameters but also could potentially be making inroads into the biomedical field with the use of selenium as a biomedical-oriented alloying element.

**Keywords:** magnesium; selenium; mechanical properties; corrosion resistance; lightweight materials; powder metallurgy





## 1. Introduction

Magnesium as a material has immense potential within the modern context; it is not only very abundant within the Earth's crust but also one of the lightest structural metals in existence [1]. This, along with world Mg primary production exceeding one million tons [2], presents a compelling case for the development of Mg materials to not only leverage its present quantities but also its qualities, such as low density for use in weight-saving engineering applications and biocompatibility for temporary implant applications.

In the past, studies on the property enhancement of Mg materials were conducted utilizing micro- and nano-sized reinforcements and alloying elements [3–7]. These studies have shown that from the viewpoint of Mg as a structural material in engineering applications, the addition of alloying elements and/or reinforcements provided significant improvements in the hardness, load bearing, and deformability of Mg materials and thus their viability as engineering materials.

Another potential application of Mg materials is in the medical sector, specifically as implants. Contemporary applications of metals in the biomedical field take the form of fasteners and supports such as plates, screws, pins, etc. [8], with materials such as steel and titanium having widespread use [9]. In this field, a different set of challenges presents itself, ranging from the need to match the property of human bone as closely as possible to avoid stress-shielding [10], minimizing the typically high corrosion rates in the human body environment [11], and overcoming concerns regarding toxicity or lack of biological incompatibility of the very materials used as implants or fasteners in the case of steels (via their alloying elements, e.g., nickel and chromium) [12]. As a metal which is not only biocompatible with, but also potentially bioresorbable within, the human body [13,14], magnesium is potentially well suited for such applications.

Selenium is a metalloid with several key functions within the human body as a biocompatible and essential trace metal, ranging from probing biological functions (biosensing) and

catalytic functions involving enzyme production to the regulation of hormone metabolism and antioxidants [15,16]. Furthermore, there is also potential for its use in the treatment of cancer (specifically clear cell renal cell carcinoma, ccRCC), provided that optimal dosage and sequence of administration are applied [17]. In addition, its anticancer properties are also derived from multiple mechanisms such as countering heavy metal toxicity, antioxidative properties, as well as the ability to inhibit the angiogenesis, migration, and invasion of certain types of cancer cells [18,19]. Furthermore, the co-supplementation of Mg and low-dose Se has shown promise in reducing liver total cholesterol levels in animal trials [20].

As an additive to biomedical materials, selenium also poses several challenges, including toxicity when ingested above a certain threshold [21]. Despite these challenges, the importance of selenium within the human body makes it an attractive frontier for the development of biomedical-oriented materials.

With these in mind, this work sets out to develop and investigate the prospect of selenium as an alloying element using magnesium as the base metal in light of the positive properties of both elements. As of writing, this study represents the first work of any kind concerning the addition of selenium as an alloying element in magnesium, thus providing an important basis on which further works and studies may be designed and conducted.

## 2. Materials and Methods

### 2.1. Materials

Table 1 outlines the raw materials used in this work. Figure 1 shows micrographs of the raw Mg (Merck Group, Darmstadt, Germany) and Se powders (Alfa Aesar GmbH & Co KG, Haverhill, MA, USA) used, as well as that of the blended Mg-15Se powder, showing their different sizes and morphologies.

**Table 1.** Raw materials used in this work.

| Raw Material | Supplier | Purity | Density (g/cc) |
| --- | --- | --- | --- |
| Magnesium Powder, 60–300 μm | Merck | ≥98.5% | 1.738 |
| Selenium Powder, 200 mesh (~74 μm) | Alfa Aesar | 99.999% | 4.28 |

The Mg and Se powders appear to be thoroughly mixed, with the larger Mg powder being more prominent in the foreground against the smaller Se particles in the background. The two powders were analyzed for size distribution. The results are presented in Table 2 and Figure 2.

**Table 2.** Experimental average sizes of Mg and Se powders used in this work.

| Raw Material | Average Powder Diameter (μm) |
| --- | --- |
| Mg powder | 254 ± 63 |
| Se powder | 28 ± 11 |

It can be seen that while the Mg powder (with its average diameter of 254 μm) conformed to the size indicated by the supplier, the Se powder was actually found to be finer in size (28 μm average diameter) compared to the information supplied (74 μm).

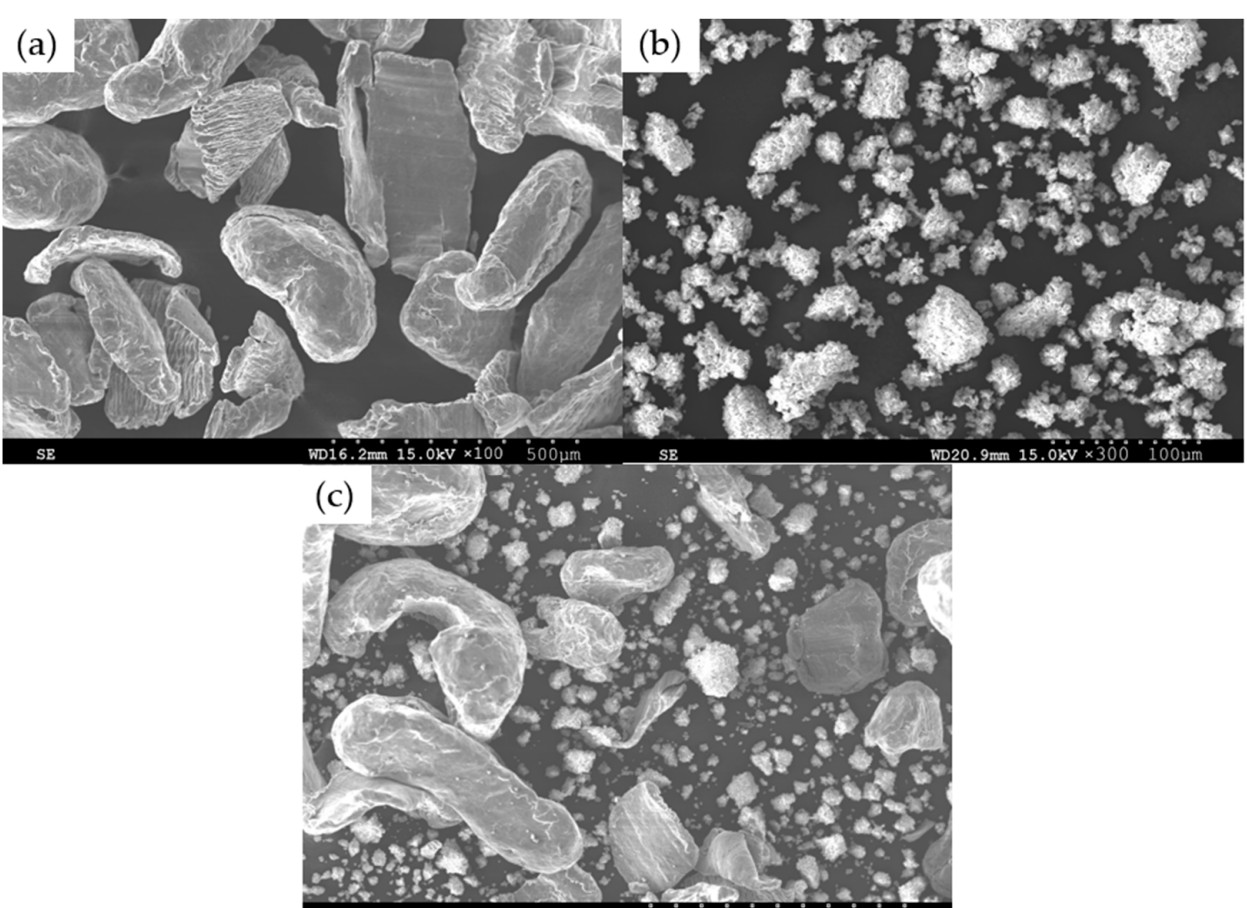

**Figure 1.** Micrographs of powder materials used in this work: (**a**) Mg, (**b**) Se, and (**c**) Mg-15Se after blending.

### 2.2. Synthesis

Mg-15Se (85 wt.% Mg and 15 wt.% Se) was synthesized using powder metallurgy by mixing the two powders in a sealed vessel using an Inversina 2 L tumbler mixer (Bioengineering AG, Wald, Switzerland) at a mixing speed of 50 rpm for a duration of 1 h. The resulting powder mixture was then compacted in a hydraulic press to a pressure of 1000 psi (6.89 MPa) with a 60 s holding time to generate a billet of 35.5 mm diameter and 45 mm height.

The billet was then sprayed with a layer of colloidal graphite and subjected to microwave sintering using a Sharp R898C(S) microwave oven rated at 900 W to a temperature of 200 °C, following which it was heated at 400 °C for 60 min and then subjected to direct hot extrusion through an 8 mm diameter die at 350 °C (corresponding to an extrusion ratio of 20.25).

### 2.3. Materials Characterisation

### 2.3.1. Microstructure

Flat and parallel samples were fine-finished using 4000 grit sandpaper, followed by polishing using alumina suspension to 0.05-micron size with Deionized (DI) water as the polishing lubricant. A Hitachi S-4300 Field Emission Scanning Electron Microscope (FESEM, Hitachi, Ltd., Tokyo, Japan) was used to obtain scanning electron micrographs, and the accompanying energy dispersive X-ray (EDX) analysis capabilities also provided material composition analysis results.

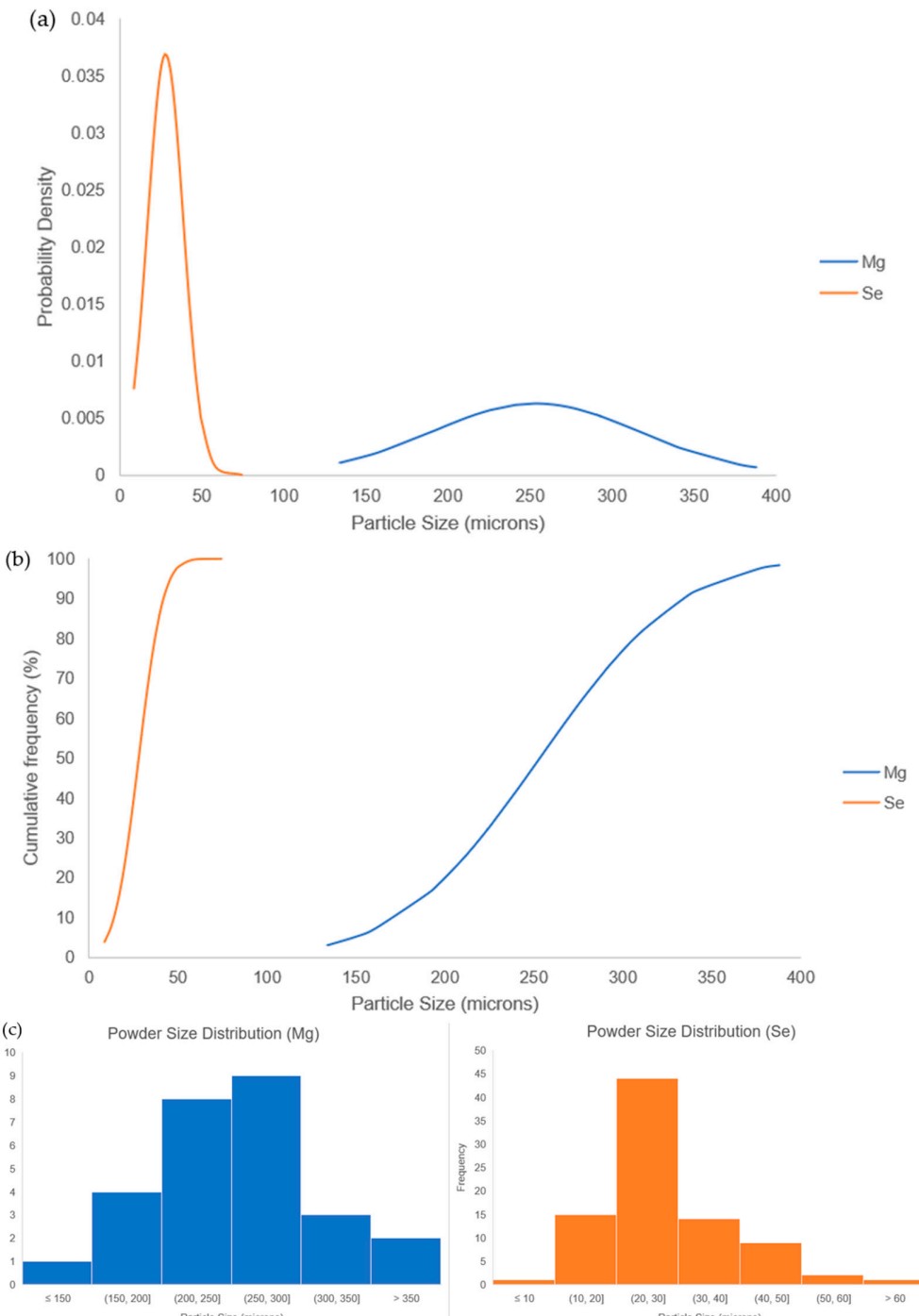

**Figure 2.** Distribution of powder sizes of Mg and Se powders used in this work in (**a**) normal distribution form, (**b**) cumulative frequency form, and (**c**) histogram form.

Grain size characterization was conducted by etching polished samples using a solution of 4.2 g citric acid, 10 mL ethylene glycol, and 100 mL DI water for a duration of 34 s. Grain images were taken using a Leica DM2500 optical microscope (Leica Microsystems (SEA) Pte Ltd., Singapore), and MATLAB (version R2013b, Natick, MA, USA) was used to conduct image analysis and grain quantification in accordance with standard ASTM E112-13(2021).

### 2.3.2. X-ray Diffraction

X-ray diffraction was conducted using a Shimadzu XRD-6000 X-ray diffractometer (Shimadzu Corporation, Kyoto, Japan) with Cu-K$\alpha$ radiation ($\lambda = 1.54056$ Å) and scanning parameters of speed, 2° per minute, and scanning angle 2θ in the range of 10° to 80°.

### 2.3.3. Density and Porosity

The density of the compacted Mg-15Se billet (both green and sintered) was calculated assuming a perfect cylinder and by mass measurement using a GH-252 electronic scale (AND Company, Limited, Tokyo, Japan).

The experimental density of the extruded Mg-15Se material was determined by the Archimedes principle, using an AD-1653 Density Determination Kit (AND Company, Limited, Tokyo, Japan) mounted on a GH-252 electronic scale (AND Company, Limited, Tokyo, Japan). A minimum of 5 samples' readings were taken.

In addition, a Hitachi S-4300 Field Emission Scanning Electron Microscope (FESEM, Hitachi, Ltd., Tokyo, Japan) was utilized to observe the presence of pores on the surface. To calculate the experimental porosity of the materials, a MATLAB program (Version R2013b, The Mathworks Inc., Natick, MA, USA) was used to measure the area fraction of the pores by dividing the total area of the detected pores by the total area of the representative sample images.

### 2.3.4. Damping Analysis

Rods of 50 mm length were cut and subjected to impulse excitation in conjunction with Response Frequency Damping Analyzer (RFDA, version 8.1.2) software (IMCE, Genk, Belgium). The resulting recorded vibration signals from the rod were then analyzed to obtain the damping properties (attenuation coefficient, damping capacity, and elastic modulus) of the material.

### 2.3.5. Mechanical Properties

A Shimadzu HMV-2 hardness tester (Shimadzu Corporation, Kyoto, Japan) was used to perform microhardness characterization in accordance with the procedures outlined in ASTM E-384 using a diamond indenter with phase angle of 136° with a load of 25 g force at a dwell time of 15 s. In total, 15 hardness measurements were taken across 1 sample surface.

Flat and parallel samples with an L/D ratio of 1 (8 mm diameter and height) were subjected to compressive load testing to failure using an MTS E-44 compressive testing machine (MTS Systems, Eden Prairie, MN, USA) with a strain rate of 0.5%/min ($8.3 \times 10^{-5}$ s$^{-1}$) in accordance with the ASTM standard E9-09. A minimum of 3 representative samples were compressed to failure to obtain the compressive properties.

Compressive fractography was also conducted using an FESEM to observe and investigate the fracture surface.

### 2.3.6. Thermal Properties

Using a Shimadzu DTG-60H Thermogravimetric Analyzer (Shimadzu Corporation, Kyoto, Japan), samples of approximately 2 mm $\times$ 2 mm $\times$ 2 mm size were subjected to an environment of 30 °C to 1400 °C at a rate of 10 °C per minute in purified air (50 mL/min flow rate) to determine the material's thermal response. The temperature reading immediately prior to a sudden spike or increase in temperature and followed by recovery, corresponding to material ignition, was taken as the ignition temperature.

Samples of the same dimensions were also subjected to an environment of 30 °C to 600 °C at a rate of 5 °C per minute in argon gas of a 25 mL/min flow rate using a Shimadzu DSC-60 Digital Scanning Calorimeter (Shimadzu Corporation, Kyoto, Japan) to measure the thermal flow of the sample with increasing sample temperature.

The coefficient of thermal expansion (CTE) was investigated on samples of 8 mm diameter and 5 mm length using a TMA PT1000 thermo-mechanical analyzer (Linseis

Messgeraete GmbH, Selb, Germany) at 50 °C to 400 °C at a 5 °C/min ramp rate in argon gas of a 0.1 L/min flow rate.

### 2.3.7. Corrosion Response

Samples of approximately 1.5 mm thickness were subjected to corrosion testing by immersion in PBS (Phosphate-Buffered Saline, Thermo Fisher Scientific Inc., Waltham, MA, USA) solution at a temperature of 37 °C for a duration of 28 days or until sample disintegration, whichever was earlier. Weight loss data were obtained in 24 h intervals by the removal of corrosion products, which was achieved by immersing the corroded samples in a solution consisting of 1.5 g $AgNO_3$ and 15 g of $CrO_3$ in 100 mL of DI water. This was followed by cleaning with DI water, and the samples were then weighed. The following formula was used to calculate the corrosion rate using the weight loss method [22]:

$$\text{Corrosion rate} = \frac{87.6 \times \text{Weight loss (mg)}}{\text{Experimental Density (g/cm}^3) \times \text{Surface Area (cm}^2) \times \text{Immersion Time (hours)}}$$

## 3. Results

### 3.1. Synthesis

The material was synthesized successfully with a solid, continuous extrudate generated with minimal surface-level cracks. However, it was observed that the sintering setup as well as the furnace used for billet soaking immediately prior to extrusion were stained red, which implies the loss of selenium content despite the presence of a colloidal graphite layer applied to the billet prior to both sintering and hot extrusion. This is likely due to the fact that the sintering temperature and extrusion temperature conditions were near and above the melting point of selenium (220 °C), respectively.

Furthermore, it was found that the resulting extruded rod has visible pores, as seen in Figure 3.

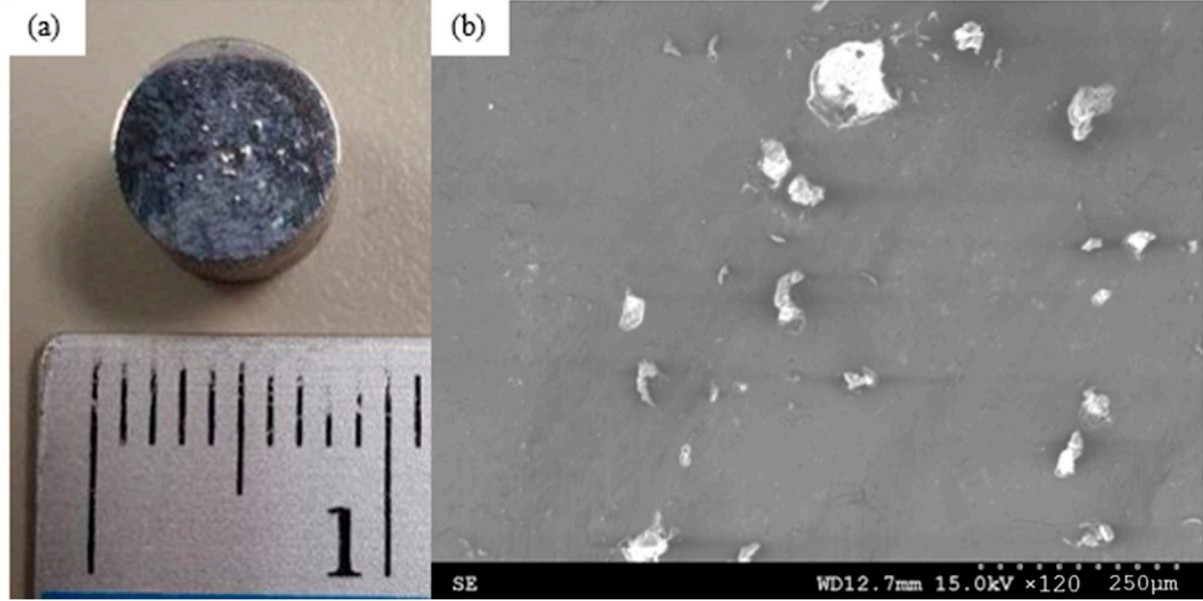

**Figure 3.** Cross-section of Mg-15Se showing visible pores on (**a**) macro-scale and (**b**) under SEM.

### 3.2. Microstructure

Figure 4 shows the microstructure of Mg-15Se, showing the presence of further small pores as well as swirls of white/bright particles against the Mg matrix.

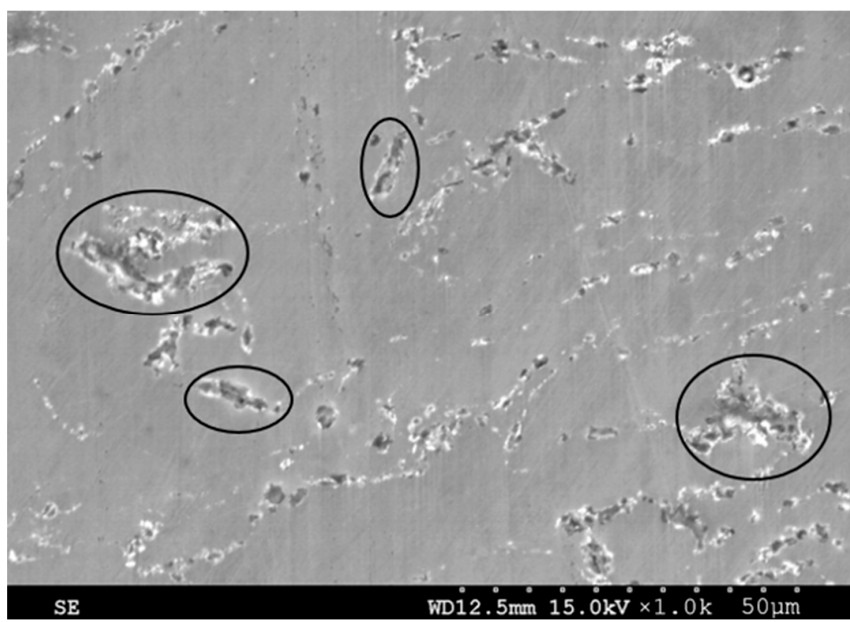

**Figure 4.** Scanning electron micrograph of Mg-15Se, with some pores circled in black.

Figure 5 shows the mapping results of Mg-15Se and selected regions for EDS analysis within the microstructure, while Table 3 shows EDS results indicating the detected elements corresponding to the various selected spectrum locations, with Se-containing regions being primarily constrained to areas with bright particles or pores, while the matrix area is predominantly Mg with some Se content distributed within.

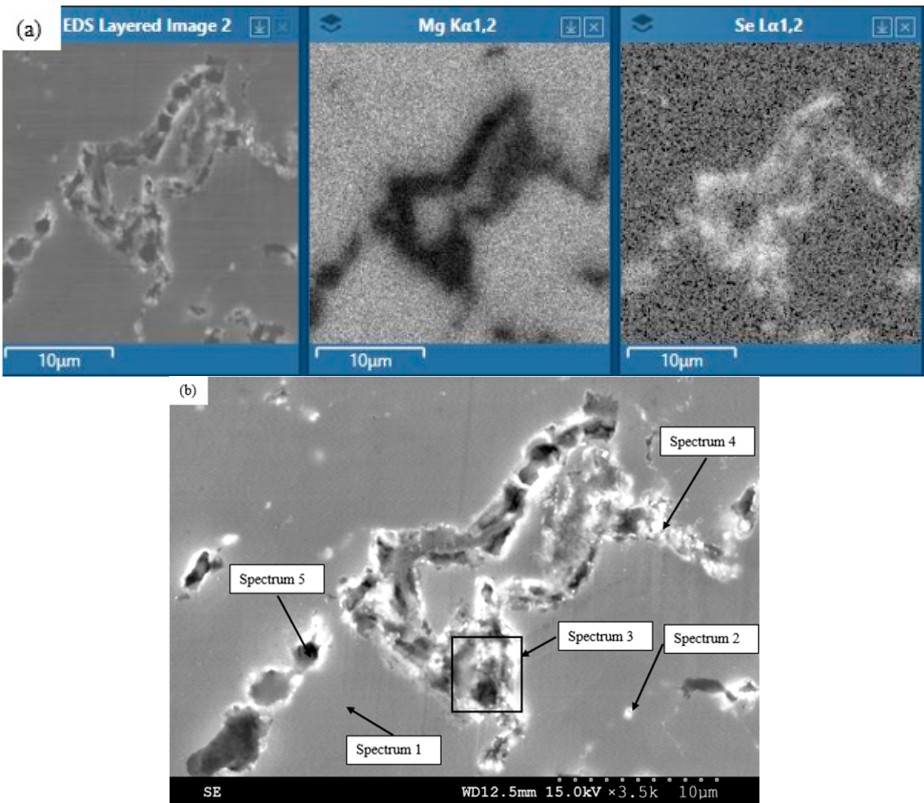

**Figure 5.** Scanning electron micrograph of representative Mg-15Se microstructure with (**a**) mapping of Mg and Se across surface and (**b**) indicated selected spectrum locations for EDS analysis.

**Table 3.** Tabulated EDS results of selected Mg-15Se spectrum regions.

| Spectrum | Detected Element (wt. %) | |
| --- | --- | --- |
| | Mg | Se |
| 1 | 99.4 | 0.6 |
| 2 | 78.9 | 21.1 |
| 3 | 53.1 | 46.9 |
| 4 | 69.1 | 30.9 |
| 5 | 79.4 | 20.6 |

The grain diameter of Mg-15Se was observed to be much lower than pure Mg, as seen in Table 4. A representative micrograph of the grains can be seen in Figure 6.

**Table 4.** Grain size characterization results of Mg-15Se.

| Material | Average Grain Diameter (µm) |
| --- | --- |
| Pure Mg [23] | 34 ± 2 |
| Mg-15Se | 9 ± 3 (↓74%) |

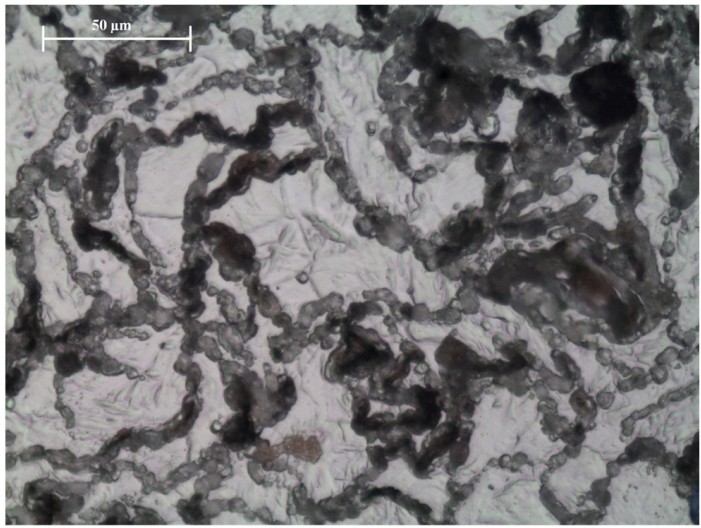

**Figure 6.** Optical of Mg-15Se material studied in this work showing the grain morphology.

### 3.3. X-ray Diffraction

The X-ray diffractogram of Mg-15Se studied in this work is shown in Figure 7. The diffractogram displayed several peaks belonging to Mg, Se, MgO, and MgSe. This was checked against the Powder Diffraction File (PDF-4+ 2023) [24,25] with card numbers 00-004-0770 (Mg), 00-051-1389 (Se), 00-004-0829 (MgO), and 01-073-6986 (MgSe). The result implies that during the synthesis, MgSe was formed.

The intensities of the Mg-15Se peaks at 2θ = 32°, 34°, and 36° (corresponding to the 10-10 prism, 0002 basal, and 10-11 pyramidal planes of magnesium, respectively) are compared with that of pure Mg, as seen in Table 5. Mg-15Se exhibited weaker prism and pyramidal texture. Mg-15Se shows a maximum peak intensity at 34°, exhibiting strong basal texture perpendicular to the extrusion direction.

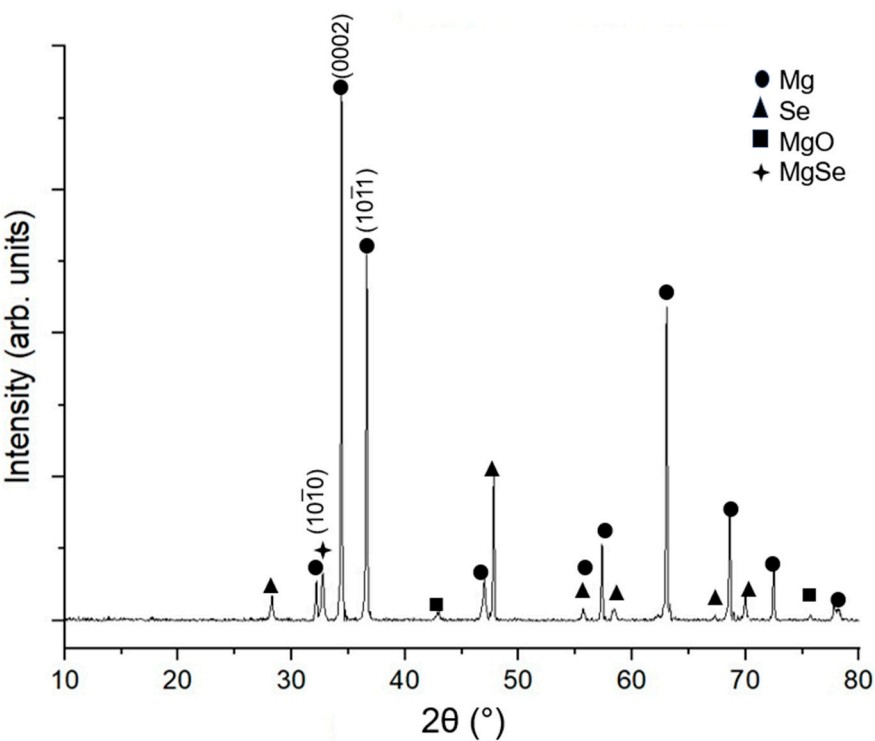

**Figure 7.** X-ray diffractogram of Mg-15Se material studied in this work.

**Table 5.** X-ray diffraction results of Mg-15Se with pure Mg as a reference.

| Material | Plane | $I/I_{max}$ |
|---|---|---|
| Pure Mg [26] | 10-10 Prism | 0.147 |
| | 0002 Basal | 1 |
| | 10-11 Pyramidal | 0.794 |
| Mg-15Se | 10-10 Prism | 0.074 |
| | 0002 Basal | 1 |
| | 10-11 Pyramidal | 0.692 |

*3.4. Density and Porosity*

The green and sintered densities of the Mg-15Se material have been outlined in Table 6, showing the very slight decrease in density just prior to extrusion.

**Table 6.** Green and sintered densities of Mg-15Se compacted billets.

| Material | Theoretical Density (g/cm$^3$) | Experimental Density (g/cm$^3$) |
|---|---|---|
| Mg-15Se, green compact | 1.9079 | 1.6962 |
| Mg-15Se, microwave sintered compact | 1.9079 | 1.6956 |

The pore area fraction of the extruded material was found to be 2.91% using image analysis. Thus, the theoretical density calculated using the rule of mixtures would have to be adjusted to account for this reduction in material across a given volume. Table 7 outlines this process:

**Table 7.** Adjustment of Mg-15Se theoretical density accounting for observed porosity within the material.

| Material | Initial Theoretical Density (g/cm$^3$) | Adjusted Theoretical Density (g/cm$^3$) | Porosity (%) |
|---|---|---|---|
| Mg-15Se | 1.908 | 1.854 | 2.91 * |

* Porosity calculated using image analysis.

Based on the results shown in Table 7, which account for the pore area fraction of the Mg-15Se material, it is then possible to calculate the retained Se content within the material using the rule of mixtures, which was found to be approximately 12.5%. This is outlined in Table 8.

**Table 8.** Theoretical and measured experimental density of Mg-15Se in this work.

| Material | Experimental Density (g/cm$^3$) | Retained Se Content (%) | Porosity (%) |
|---|---|---|---|
| Pure Mg [22] | 1.736 | - | 0.21 |
| Mg-15Se | 1.823 | 12.5 | 2.91 * |

* Porosity calculated using image analysis.

### 3.5. Damping Analysis

The vibration response curve plotted as part of the damping analysis in Figure 8 was prescribed an exponential best-fit curve with a general form of $Ae^{-bt}$, where b is the derived attenuation coefficient. The results indicated that Mg-15Se exhibited a superior damping capacity, as outlined in Table 9, with only a minor reduction in Young's modulus compared to pure Mg.

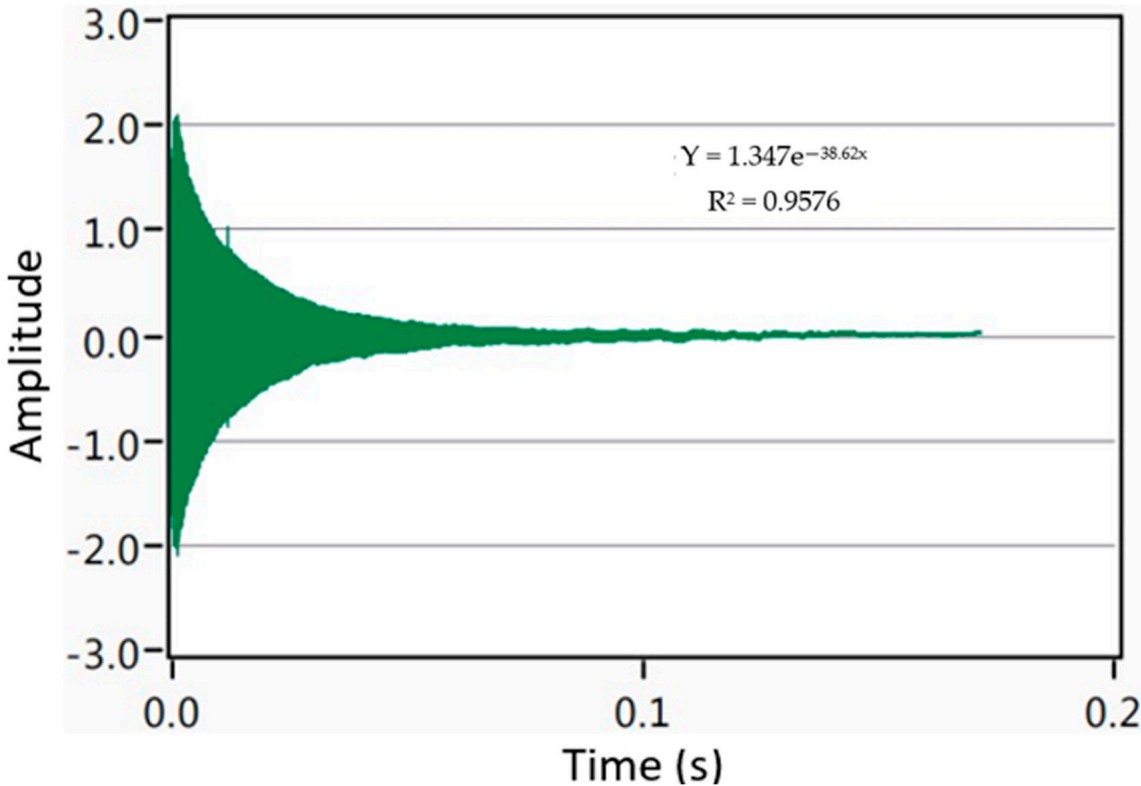

**Figure 8.** Vibration response curve of Mg-15Se with best-fit curve equation.

**Table 9.** Damping properties and Young's modulus of samples.

| Material | Attenuation Coefficient | Damping Capacity | Young's Modulus (GPa) |
|---|---|---|---|
| Pure Mg [23] | - | 0.000656 | 44.7 |
| Mg-15Se | 38.62 | 0.001155 (↑76%) | 42.9 (↓4%) |

### 3.6. Mechanical Properties

3.6.1. Hardness

The microhardness values of Mg-15Se studied in this work are presented in Table 10, showing an increase in hardness relative to pure Mg.

**Table 10.** Average microhardness values of Mg-15Se compared to pure Mg.

| Material | Average Microhardness (HV) |
|---|---|
| Pure Mg [23] | 46 ± 3 |
| Mg-15Se | 72 ± 5 (↑57%) |

3.6.2. Compressive Properties

The compressive test results of Mg-15Se are compiled in Table 11, with the stress–strain curve outlined in Figure 9. Significant increases in yield strength, ultimate compressive strength, fracture strain, and work of fracture were all observed with respect to the Mg-15Se material over pure Mg.

**Table 11.** Compressive properties of Mg-15Se studied in this work with Pure Mg as a reference.

| Material | 0.2% Compressive Yield Strength (MPa) | Ultimate Compressive Strength (MPa) | Fracture Strain (%) | Energy Absorbed (MJ/m$^3$) |
|---|---|---|---|---|
| Pure Mg [23] | 72 ± 5 | 174 ± 7 | 16 ± 2 | 23 ± 2 |
| Mg-15Se | 87 ± 3 (↑21%) | 263 ± 12 (↑51%) | 24 ± 2 (↑50%) | 39 ± 6 (↑70%) |

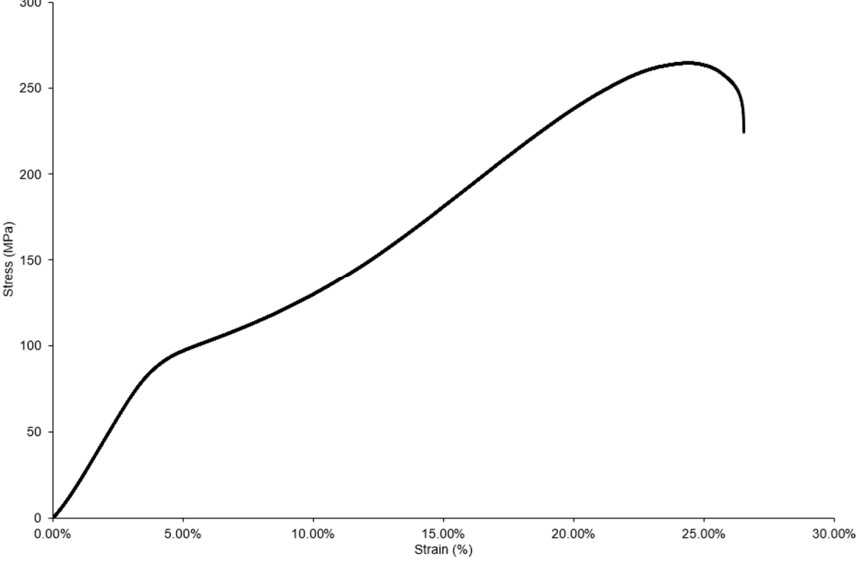

**Figure 9.** Representative stress–strain curve of Mg-15Se material.

Figure 10 shows the resulting microstructure of the fractured samples, showing the 45-degree fracture angle and the resulting shear band morphology, as well as the presence of cracks on the fractographs.

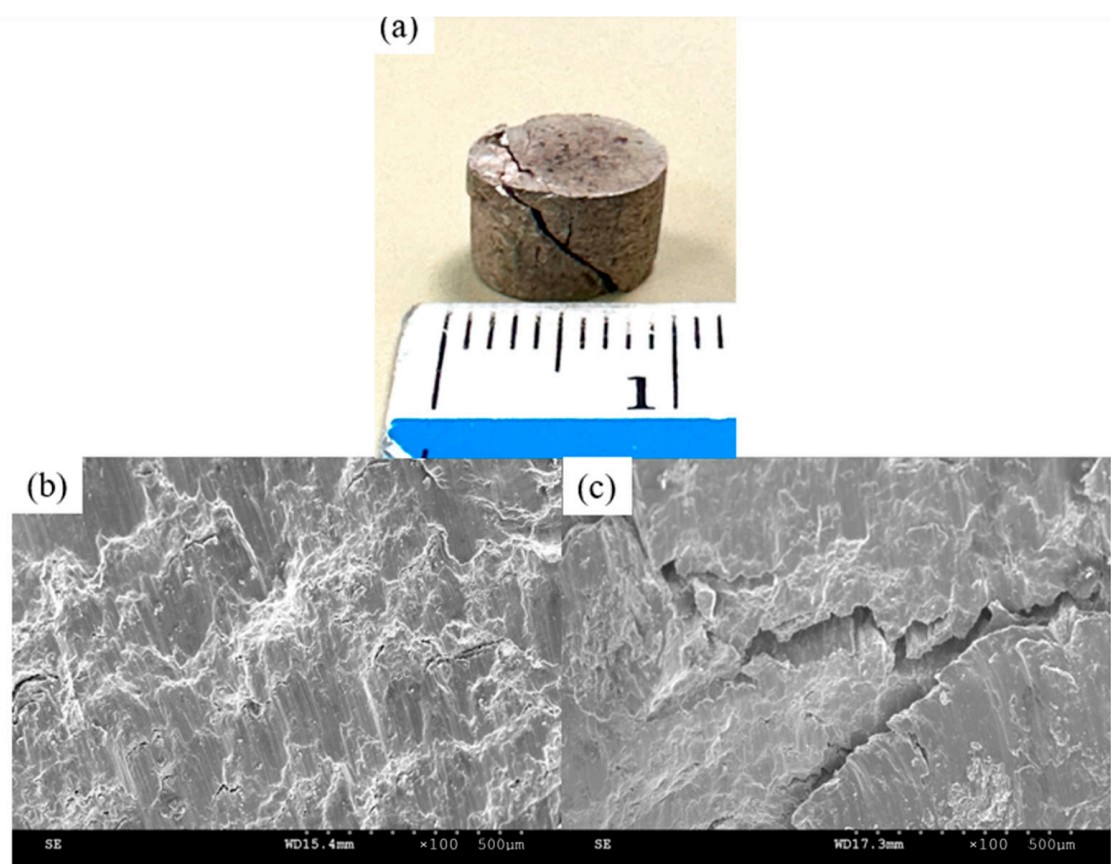

**Figure 10.** Fractography of Mg-15Se in this work: (**a**) macro-scale photograph showing fracture line and (**b**,**c**) micro-scale fracture surfaces showing shear bands and cracks respectively.

### 3.7. Thermal Properties

Table 12 shows the ignition temperature results of the TGA testing. The results are compared with pure Mg, and it was found that there was a 64 °C increase in ignition temperature.

**Table 12.** Ignition temperature of Mg-15Se with pure Mg as a reference.

| Material | Ignition Temperature (°C) |
| :---: | :---: |
| Pure Mg [26] | 581 |
| Mg-15Se | 645.5 ± 1.5 (↑ 11.1%) |

Figure 11 displays the DSC response curve of Mg-15Se.

Table 13 shows the coefficient of the thermal expansion of Mg-15Se, which is slightly reduced when compared to pure Mg.

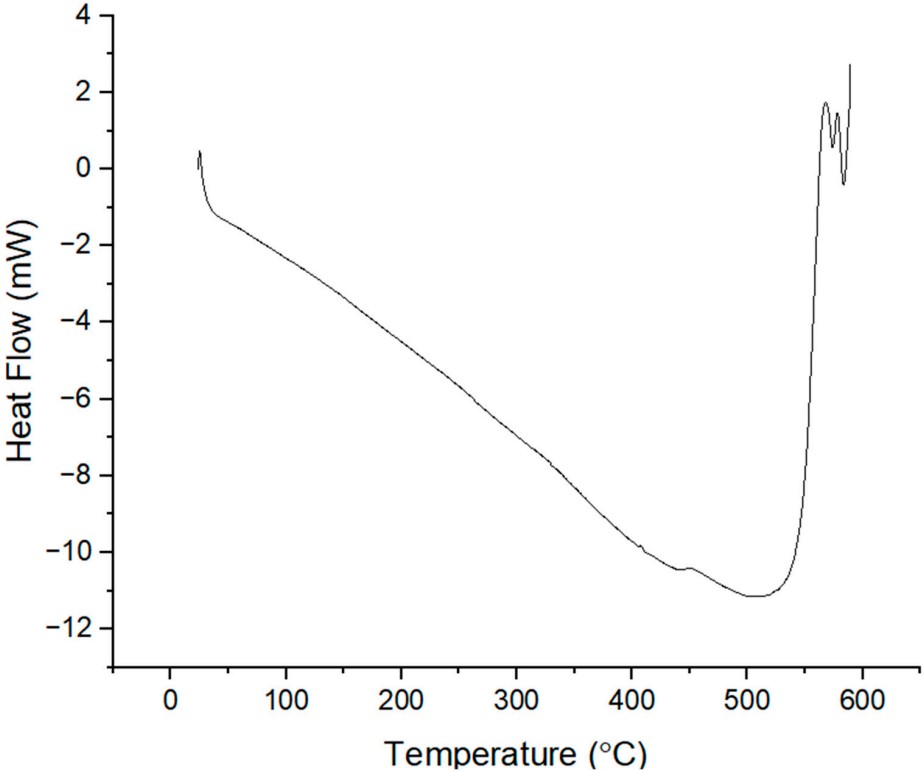

**Figure 11.** Plot of heat flow vs. temperature of Mg-15Se.

**Table 13.** Coefficient of thermal expansion of Mg-15Se with pure Mg as a reference.

| Material | Average Coefficient of Thermal Expansion ($\times 10^{-6}$/K) |
|---|---|
| Pure Mg [26] | 25.3 |
| Mg-15Se | 24.2 ± 0.5 (↓4.8%) |

### 3.8. Corrosion Response

Figure 12 shows the corrosion rates of Mg-15Se with exposure duration. It was found that the samples would last, on average, 16 days in the corrosive environment before disintegrating. Table 14 shows the average corrosion rate of Mg-15Se.

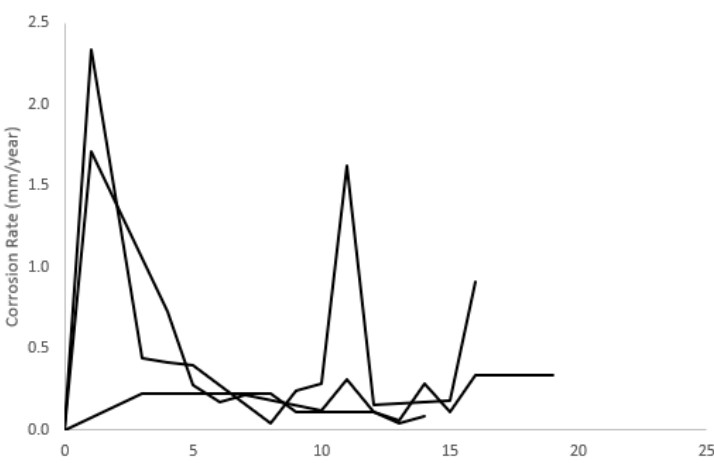

**Figure 12.** Corrosion rates of Mg-15Se in this work, with each line representing one sample.

**Table 14.** Corrosion rate of Mg-15Se with other pure Mg materials from the literature as a reference.

| Material | Average Corrosion Rate (mm/Year) |
| --- | --- |
| Pure Mg, extruded and cold drawn [27] | 2.2 |
| Pure Mg, extruded [27] [28] | 1.2 |
| Pure Mg, porous (scaffolds) [29] | 0.5–3.7 |
| AZ91, powder metallurgy [27,30] | 1.6 |
| AZ91-0.5CNT (vol.%), powder metallurgy [30] | 12.5 |
| AZ91-0.5Graphene (vol.%), powder metallurgy [30] | 3.5 |
| AZ91-0.5Fullerine (vol.%), powder metallurgy [30] | 2.2 |
| Mg-15Se (current study) | 1.5 ± 1.1 |

## 4. Discussion

### 4.1. Synthesis

While Mg-15Se was successfully synthesized, there was Se loss, which likely occurred during exposure to elevated temperatures in processing, specifically during pre-extrusion heating to 400 °C, which is beyond the melting point of selenium. In addition, the presence of porosity of nearly 3% despite sintering and an extrusion ratio of 20.25 implies that the high Se volume fraction contributes to increased porosity, suggesting that the synthesis parameters could be further optimized for future works, including more refined blending for increased homogeneity, as well as a reduction in Se loss during the extrusion stage, where the temperature was beyond the melting point of selenium.

### 4.2. Microstructure

The resulting pores and observation that selenium was detected mainly near/at the pores indicate the loss of selenium primary during the soaking/extrusion stage. This increased porosity was accompanied by a matrix, which appears to be continuous, implying that the matrix portion of the material is otherwise bonded well.

The significant degree of grain refinement (by 74% compared to pure Mg) can be attributed to Particle-Stimulated Nucleation (PSN) on the part of micro-scale selenium used as the alloying element and MgSe phase [31,32]. As selenium peak was observed in XRD studies (Figure 4), the present material can be classified as an alloy–composite. Note that the Mg-Se binary phase diagram has not yet been investigated and does not exist in the open literature.

### 4.3. X-ray Diffraction

The X-ray results indicate that apart from Mg and Se, small amounts of MgO and MgSe were also detected. Due to the blending of powder as well as compaction of the billet taking place without the use of protective/inert gas, some oxidation of the magnesium powder resulting in MgO can be expected as a result of the processing method. For MgSe, this intermetallic phase was previously found to be present at room temperature in a past work by Broch [33]. Given that the powder blending was conducted, as well as pre-extrusion heating, at 400 °C (beyond the melting point of selenium of 221 °C), it is likely that this intermetallic phase was present across the surface of the material after precipitation, though it is difficult to ascertain/isolate this phase specifically using SEM or EDS characterization.

With regard to the texture, the strong basal texture which was observed in the longitudinal direction is consistent with that of pure Mg and its alloys [34], unlike with the addition of nano-scale ceramics, as previously investigated by Parande et al. [26], suggesting that the addition of Se alters the texture of Mg in a different way compared to other composite reinforcements.

### 4.4. Density and Porosity

It was found that the retained Se content within the material was approximately 12.5%. This was derived from the observed pore area fraction, which was 2.91%, which equates to the true porosity of the material and that above pure Mg (0.21%). A possible

reason for this, apart from the use of Se as an alloying element, is the lower sintering temperature; the material in this work was sintered to just 200 °C (due to concerns of Se melting) compared to the reference material, which was sintered to 640 °C. A higher sintering temperature/duration was previously observed to have resulted in narrower and smaller pores in both Mg and non-Mg materials [35–37], likely as a result of more interconnected particles as sintering temperature increases. Another reason for the high porosity of the Mg-Se alloy can be attributed to the partial leaching/melting of selenium during the extrusion step, as indicated in Section 4.2.

### 4.5. Damping Analysis

The damping analysis indicates that Mg-15Se possesses superior damping properties when compared to pure Mg, notably damping capacity. In addition, Young's modulus was also slightly decreased to 42.9 GPa, which can be attributed to the presence of selenium, which has a lower Young's modulus of 10 GPa [38]. This represents an improvement from a biomedical standpoint; the human cortical bone has a Young's modulus of 19 to 22.5 GPa [39,40], and thus Mg-15Se is closer to the human bone in this property (property-matching).

With regard to the improvement in damping capacity, this can be attributed to the increased presence of pores within the material as a natural consequence of the powder metallurgy method used; these act as sites of energy conversion (via deformation and displacement) from kinetic energy (from vibrations, etc.) to thermal energy, leading to their dissipation. This beneficial effect of pores and voids has been observed in past works involving the intentional synthesis of porous Mg materials [41,42].

### 4.6. Mechanical Properties

#### 4.6.1. Hardness

The increase in room temperature hardness of Mg-15Se can be explained by the Hall–Petch relationship [43], indicated by the much smaller average grain diameter when compared to pure Mg. This decrease in grain diameter can be attributed to the presence of secondary phases such as Se, MgSe, and MgO in the matrix (Figure 4). Further, the secondary phases with different intrinsic CTE when compared to Mg lead to an increase in dislocation density, thus making localized deformation more difficult.

#### 4.6.2. Compressive Properties

Substantial increases in compressive yield strength (21%), ultimate compressive strength (51%), fracture strain (50%), and work of fracture (70%) were observed, which can be attributed to the strengthening mechanisms associated with the presence of secondary phases as well as grain refinement, as observed from grain size characterization in accordance with the Hall–Petch relationship [44–47]. Furthermore, as mentioned earlier in Section 4.2, grain refinement was also achieved through PSN, the benefits of which were sufficient to enhance the overall compressive properties, including fracture strain despite the higher relative strength of the basal texture exhibited by Mg-15Se from X-ray analysis.

The resulting fracture surface was found to be In line with that of other similarly processed Mg materials [23], though with deeper cracks on the fracture surface, which are similar to those found in porous Mg-SiO$_2$ nanocomposites [35], suggesting a possible link between the presence of these cracks and the relatively high porosity of the Mg-15Se studied.

### 4.7. Thermal Properties

The ignition temperature of Mg-15Se was observed to be 11% higher than pure Mg at 645 °C. While increased ignition resistance is not significant in the context of biomedical applications, this may provide some potential for the further use of selenium in the alloying of other Mg materials as ignition resistance has historically been a compromise in Mg materials.

The DSC results indicate a strongly exothermic thermal flow starting at approximately 540 °C and peaking just above 550 °C and remaining almost entirely exothermic past that temperature. This can be attributed to the formation of MgSe in an inert environment [48], which was estimated by Mills [49,50] to have an enthalpy of formation of $-293$ kJ/mol.

Mg-15Se was found to have a slightly lower CTE compared to pure Mg despite selenium having a higher CTE of $37 \times 10^{-6}$/K [47]. This can be attributed to the presence of secondary phases (Figure 4) which outplayed the role of higher porosity. The results thus indicate the capability of Se addition to increase the dimensional stability of magnesium.

### 4.8. Corrosion Response

The corrosion rates of Mg-15Se in this work were comparable with those of pure Mg in the past literature using a variety of processing methods (extrusion, casting, etc.), as well as being more corrosion-resistant than AZ91 alloy with carbon addition using powder metallurgy [27–30]. This is significant as the pure Mg materials were processed by both liquid-to-solid (casting) processing methods, as well as solid-to-solid (scaffold) processing methods, highlighting the feasibility of powder metallurgy, while the Mg-15Se material is resistant to corrosion relative to carbon-reinforced AZ91.

The corrosion rate exhibited by Mg-15Se despite the presence of pores and a pore area fraction (in turn, porosity) of nearly 3% can be explained by the Pilling–Bedworth Ratio (PBR) of selenium (1.69) [51], which is within the range required for protective oxide layers. This would explain the brief spikes/increases in corrosion rates with immersion duration, but this would then decrease to consistently low values as the Se becomes the next to be subject to corrosion, lowering the corrosion rate for that specific time interval.

This highlights the potential for applications of selenium addition for use in corrosive environments, which has historically been a challenge for Mg-based materials. The high corrosion resistance of selenium, implied by its PBR of 1.69, also synergizes with a potential use of Se-containing Mg materials in the biomedical field, where the increased corrosion resistance conferred would result in a longer useful lifespan of any Mg-based implant. In addition, it was demonstrated that mechanical enhancements were exhibited without compromising corrosion response.

### 5. Conclusions

In this study, Mg-15Se was successfully synthesized with powder metallurgy and, for the first time, investigated for the effect of selenium addition as an alloying element on Mg, with the following conclusions being observed:

1. The resulting Mg-15Se material had a higher porosity (2.91%) when compared to pure magnesium (0.21%). This can be attributed to the partial loss of selenium primarily during the extrusion stage.
2. The resulting Mg-15Se material exhibited a relatively stronger basal texture than that of pure Mg, and it also underwent significant grain refinement (74% smaller grain diameter than pure Mg).
3. Mg-15Se had a superior damping capacity to pure Mg (76% increase).
4. Mg-15Se had superior hardness (57% increase) as well as compression properties (21% increase for 0.2% yield strength, at least 50% increases for ultimate compressive strength, failure strain, and energy absorbed).
5. The average corrosion rate for Mg-15Se is comparable with extruded pure Mg from multiple works, despite its high porosity, due to the high PBR of Se at 1.69, showcasing the lack of compromise in corrosion resistance whilst possessing a superior mechanical response.
6. Thermal analysis also indicated that Mg-15Se is more ignition-resistant than pure Mg despite its much higher porosity.
7. In this study, it was discovered that Se loss occurred during the extrusion stage; additional steps to reduce Se loss by use of a lower temperature in processing or

further protective/containment measures is possible, but this would have to be studied to see if there is an adverse effect on the final material.

In essence, considering the potential of Se as a biomedical-oriented element, the prospects of subsequent studies go beyond merely structural applications; as the use of Mg materials in the biomedical field increases, selenium can be utilized as an alloying element without undue concerns of degradation in neither mechanical nor corrosion response. In addition, further refinements for Mg-Se materials (in biomedical contexts, e.g., bioresorbable implants) would include controlling the corrosion rate of the material as well as the rate of Se release/dosage to ensure both the good functioning/structural integrity of the implant as well as the nutritional/supplemental nature of Se within medically safe ranges.

**Author Contributions:** Conceptualization, M.G.; methodology, M.J. and M.G.; validation, M.J.; formal analysis, M.J. and V.S.; investigation, V.S.; resources, M.J. and M.G.; data curation, M.J. and V.S.; writing—original draft preparation, M.J.; writing—review and editing, M.J. and M.G.; visualization, M.J. and V.S.; supervision, M.G.; project administration, M.J. and M.G. All authors have read and agreed to the published version of the manuscript.

**Funding:** This research received no external funding.

**Data Availability Statement:** The original contributions presented in the study are included in the article, further inquiries can be directed to the corresponding author.

**Acknowledgments:** The authors acknowledge Juraimi Bin Madon for assistance with extrusion and Ng Hong Wei for assistance with TGA, DSC, and CTE testing.

**Conflicts of Interest:** The authors declare no conflicts of interest.

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
