# Peer review of "A First-Time Investigation into Ecofriendly and Biocompatible Mg-Se Binary System for a Greener Earth"

_metals, doi:10.3390/met14020163_

Round 1

Reviewer 1 Report

Comments and Suggestions for Authors

Authors investigated the Ecofriendly and Biocompatible Mg Se Binary System For Greener Earth. This paper can be accepted after major revision.

·        What would be properties if the Se content in increased after 15%

·        How authors claim this materials as eco-friendly?

·        What is the exact application of this particular material?

·        Include some quantitative results in the abstract.

·        Mg 15Se (85 wt.% Mg and 15 wt.% Se) was synthesized using powder metallurgy- SEM morphology of the Mg and Se powders have to be included to understand the shape of the powders used in this work.

·        ‘Particle size analysis has to be reported

·        Include the details about hot extrusion process employed in this work

·        SEM images provided in this manuscript is as sintered or hot extruded?

·        Green density and Sintered density should be included

·        SEM images of the mixed powders is required to see the uniform mixture of the both powders

·        Figure 1. Cross section of Mg 15Se showing visible pores on & Figure 2.

·        Scanning electron micrograph of Mg 15Se, with some pores circled in black  - What about the Se and Grain boundaries

·        Figure 4. X Ray diffractogram of Mg 15Se material studied in this work

o   why the planes are not marked in the XRD pattern?

·        In this study, Mg 15Se was successfully synthesized with powder metallurgy – why 15 Se was selected in particular?

·        Why PM process was employed to synthesize this material?

·        Need to include stress strain plot for the compression test

Reviewer 2 Report

Comments and Suggestions for Authors

Esteemed Authors,

Your research of alloys in Mg-Se system and directing it towards medical applications is promissing, unfortunately the current state of paper needs drastic improvements, in my opinion.

In my opinion the introduction should be directed towards the physical metallurgy of magnesium and your rationale why selenium was chosen, its expected influence on structure and properties - the current knowledge for this system, not general aspects regarding magensium and selenium.

The materials and methods part must be improved, eliminating the obvious statements such as "a chart of intensity against 2theta ....". The use of a home made program for analysis needs some explanations: the method used for quantification, the standard/methodology used as reference - information that was not provided.

The results are lacking. You speak of average grain size, yet no micrographs are presented to observe the grains, same observation for porosity - higher magnification micrographs are advisable to justify your results. It would be interesting to clasify the pores.

The chemical composition determined by EDS should be presented in %wt (not in %at), it would facilitate the understanding on the distribution of Se in the structure.

In the XRD the formation presence and formation of MgSe is questionable if only one peak is present, a more thorough investigation needs to be performed and related to findings by EDS and microstructure analysis.

Please use the conventional presentation for the crystallographic planes.

The compressive stress - strain curves would provide a great insight regarding material behaviour, I strongly advise introductng them in the paper. Also the fractographic analysis needs some discussion.

The DSC curve is presented and no discussion is presented.

Please check spelling, several errors (ex. line 148 - surficial) are present.

I hope that these suggestions are helpful.

My best regards.

Comments on the Quality of English Language

English is fine, minor spelling errors are present.

Round 2

Reviewer 1 Report

Comments and Suggestions for Authors

The revised paper entitled "A First Time Investigation into Ecofriendly and Biocompatible 1 Mg-Se Binary System For Greener Earth" is recommended for publication. 

Reviewer 2 Report

Comments and Suggestions for Authors

Dear Authors,

In a large extent I see improvement in your work. My starting observations towards predicting the phases that could form in a Mg-Se alloy and, inherently, its properties, were somewhat inqusitive to see if a thermodynamical calculation of the phase diagram was attempted, and, if so, a discussion would enhance the study. Indeed, the discussion would be, in a large extent, speculative, but somehow, a verification of the predictions would have been performed given your alloy. I am fully aware of the implication of phase diagram prediction, thus I do not object that these aspects were not fulfilled. It is, in its essence, a pioneering study.

All that I mentioned aspects were corrected, my only comments are directed towards Fig. 2. If the data is vailable, I would suggest presenting the powder size distribution as a histogram and the cumulative one. Also the axis labels need to be changed, particle size vs. percent.  

My best regards.
